# How Has China Structured Its Ecological Governance Policy System?—A Case from Fujian Province

**DOI:** 10.3390/ijerph19148627

**Published:** 2022-07-15

**Authors:** Xiaojun Zhang, Weiqiao Wang, Yunan Bai, Yong Ye

**Affiliations:** 1School of Economic and Management, Fuzhou University, Fuzhou 350108, China; xiaojun@fzu.edu.cn (X.Z.); 200720193@fzu.edu.cn (W.W.); 072007204@fzu.edu.cn (Y.B.); 2School of Government, Nanjing University, Nanjing 210023, China

**Keywords:** ecological civilization, bibliometric analysis, environmental governance

## Abstract

Ecological civilization (EC) has been seen as the final goal of social and environmental reform within a given society. Much attention has been paid to the national governmental level in previous studies, and district- and local-level government actions have been a lower priority, which may have led to overlooking key details of management institutions and policy systems in relation to EC. In this research, we aimed to make a significant contribution to the literature by tracing the EC trajectory and policy transitions. Through bibliometric analysis of policy documents, we reviewed the EC construction system for Fujian Province from 2004 to 2020. The policy priorities, organization-functional network, and contributing factors to policy changes in each of the three phases are discussed in depth. Target setting, actors’ functions, and institutional guarantees are the core elements of EC construction. This research provides a quantitative foundation for understanding policy reform and transition with regards to Chinese local governments’ EC actions. The experience of Fujian Province shows trends toward legalization, multi-actor linkage, and issue refinement that may serve as a basis for other countries and regions in order to explore the promotion of sustainable development and environmental governance as pathways to EC.

## 1. Introduction

Through rapid industrialization and urbanization, China has achieved extensive economic growth and social development that have enriched all aspects of people’s lives. Meanwhile, China has been facing severe environmental problems such as pollution, ecological destruction and resource depletion [1] owing to part of its development mode, which is characterized by high pollution, high consumption, and high emissions [2]. To support the country’s long-term progress, ecological impacts must be considered, as highlighted in the national development strategy [3]. As the world is paying close attention to ecological conservation [4], China has put forward a concept called ecological civilization (EC) to manage the relationship between humans and nature comprehensively, scientifically, and systematically [5].

Since the 1990s, China has integrated sustainable development strategies into its socioeconomic development [6]. In 2012, the 18th National Congress of the Communist Party of China (CPC) creatively incorporated EC construction into the deployment of the national development plan, a “five-in-one” blueprint that included economic, political, cultural, social, and EC goals. Then, the 13th National People’s Congress included EC construction in the Constitution. Currently, EC as an ideological framework impacts the government’s policies and law making and is significantly embedded in Chinese society and even in international strategies [7,8,9]. Today, China’s EC shows great success in both concept exploration and the concrete practices of sustainable development [10]. By 2020, the country had reduced its carbon emissions by 48.4% per unit of GDP from the 2005 level [11].

The United Nations Environment Programme (UNEP) examined China’s EC efforts in 2016 and claimed that this approach may serve as a new global environmental governance system to benefit people worldwide [5]. EC has entered the vision of many sectors of society, and its influence is expanding globally [12]. To some extent, EC can also be seen as an alternative to hegemonic culture and a major path of global development [13]. Since the signing of the Paris Agreement and the initiation of the 2030 Agenda for Sustainable Development Goals (SDGs), humankind has moved into a new era of sustainable and environmentally friendly ways to survive and develop. In addition, the Decade of Restoration (from 2021 to 2030), led by UNEP and the Food and Agriculture Organization of the United Nations (FAO), was proclaimed in 2019 to call for the protection and revival of ecosystems around the world [14]. For countries worldwide, especially developing countries, much support is needed to achieve these goals [15]. Research on EC, as a new strategy that differs from previous global sustainability strategies, can be critical in contributing to global sustainability [16]. In fact, in response to environmental and ecological problems, different regions have explored different paths to solve them. For example, keeping pace with UN SDGs, Korea is transforming the goal systems of local government into those of sustainable development, and countries in Africa are localizing the sustainable development goals [17,18]. How is the Chinese government attempting to achieve sustainability, and what inspiration can the Chinese experience offer to other countries and districts?

In previous studies, scholars have focused mainly on efficiency evaluation. They have attempted to assess the practice of EC, while the government’s actions have been deemed an independent variable that affects the process [19,20,21]. In specific environmental sectors, based on EC construction essentials and connotations, some studies have created index systems to evaluate EC development levels with different indicators that are applicable to specified environmental issues, such as urban space [22], hybrid energy systems [23], industrial pollution [24], and water ecosystems [25]. The indicators employed were taken from Chinese national and local standards combined with regional features and issue characteristics. Each EC program has environmental goals that are only part of the overall EC goals; thus, it is important to evaluate them together [26]. From a holistic perspective, scholars have constructed metrics to assess EC construction effects with people-oriented indicators [27], concept-based indicators [12], literature-focused indicators [28], province-oriented indicators [29] and others. Overall EC goals are also applied to coupling evaluation and explaining cross-disciplinary topics. In addition to the coupling relationship between EC and ecotourism [30], EC development is coordinated with public governance [31], city planning [32] and urbanization development [33]. It has even been employed to clarify accounting issues [34] and issues related to the Olympic Games [35]. EC has seemingly become a key to understanding Chinese environmental governance and explaining many Chinese issues.

However, understanding EC at the level of effect evaluation is too superficial, so more and more studies focus on the concept of EC itself to trace its origins and deconstruct its connotations. Some scholars have investigated the transition process from “environmental protection” to “ecological civilization” and tried to find the traditional roots of EC [8,36,37]. To expand the definition, researchers in political science have sought more in-depth and accurate connotations of EC. Arran Gare investigated EC as a socialist construct [13], while Liu et al. explored EC from the perspective of post-socialist transitions [38]. Marinenelli Maurizio compared the concepts of the Anthropocene and EC and suggested that their similarity has a discursive power to drive the shift from binary political economy discourses to sustainability discourses [39]. When EC is considered as an ideological product, China has carried out comprehensive publicity, especially political propaganda, to influence citizens’ behavior and promote environmental protection [40,41]. However, a large amount of political research is not necessarily good for EC research. Goron Coraline performed a comprehensive review of Chinese academic and political publications to reveal the political and theoretical meanings of EC and pointed to the limitations of political discourse [42]. Ignoring the purely political attributes of EC, many EC strategies have depended on the direction of governance and reform within China [43], so some studies have focused on practical actions. As policies can be seen as government actions, some studies have paid attention to ecological redlines [44], marine protection [6], green space planning [45], pollution [46] and other environmental issues from the perspective of public policy. They have tried to elucidate a concrete environmental policy concerning the historical origins and development of EC and to explain policy effects and mechanisms with EC in the background. In addition, much attention has been paid to the national governmental level, causing a lack in district prominence and assigning lower priority to local-level government action. Ultimately, previous studies can simply be divided into three types: connotation and logic analysis, which overemphasizes theory and politics and weakens real practice; assessment of construction effectiveness, which focuses on results instead of processes; and policy transformation tracing, which focuses mainly on specific environmental sectors. Furthermore, although quantitative methods have been widely used for effect evaluation studies, process and transition analyses have not favored this approach. Therefore, can we study EC from an overall perspective and highlight the process of actual actions with quantitative methods?

This paper took a holistic view of EC as an integral policy framework instead of focusing on a specific environmental policy. Policy priorities, organization–functional networks, and the contributing factors to policy changes are discussed in depth based on the bibliometric analysis of policy documents. We hope that our findings will contribute to environmental management in other countries.

## 2. Materials and Methods

### 2.1. Data Sources and Preprocessing

Fujian Province is an important area for implementing and practicing the Xi Jinping school of thought regarding EC. Fujian Province is where his ideas on EC were first formed. In 2000, Xi Jinping, then forward-looking governor of Fujian Province, proposed the strategic concept of building an ecological province. Two years later, he formally proposed the strategy of building an ecological province in a government work report, which started the noteworthy practice of EC construction in Fujian [47]. After he became the President of China, he visited Fujian several times to conduct research on the construction of EC. In 2016, Fujian Province was listed as the country’s first national EC test area. In 2020, the *National Ecological Civilization Pilot Zone (Fujian) Implementation Program* was completed, and the reform initiatives and experience practices of Fujian Province were replicated and promoted across the whole country, which means that Fujian Province became a model zone for the construction of ecological civilization in the country. President Xi Jinping said, “Ecology is the most competitive advantage of Fujian Province”. The Xinhua News Agency [48], the official news agency of the Chinese government, stated, “In the evaluation report by the Chinese Academy of Engineering, the ecological civilization construction in Fujian province is at a leading level in China. Fujian Province has built a complete ecological civilization institution system with clear property rights, multiple participation, balanced incentives and constraints”. Until now, Fujian Province ranks first in the country in forest coverage and EC index. It can be said that Fujian is the first region in China to be influenced by Xi Jinping’s thought regarding EC and to building it most successfully. Therefore, we chose Fujian as our research object.

The policy document data investigated in this paper were derived from government official websites and the PKULaw Database (the largest Chinese policy database, which is continuously updated with the most recent documents). We used the keywords “ecological civilization” (“生态文明”) to search the database and retrieved a total of 2491 policy documents. EC first appeared in the *Outline of the Overall Plan for Ecological Province Construction*, published by the Fujian People’s Government in 2004. As noted in this document, Fujian Province planned to finish the construction of the ecological province before the end of 2020. Therefore, the retrieval time range was determined to be from 1 January 2004 to 31 December 2020. We have manually reviewed the policy texts, eliminated the texts that are not highly relevant to EC, and removed the contents that are not relevant to EC from some policy texts that are related to EC construction. After this, 1497 policy documents remained, including 699 documents at the provincial government level (i.e., those by provincial CPC, the provincial government and its direct affiliates, provincial bureaus, provincial associations, and provincial offices directly under the provincial CPC) and 798 documents at the prefecture-municipal government level. These documents include laws, administrative regulations, departmental regulations, departmental normative documents, departmental working documents and approvals. Each document included keywords, the policy-making department, level of authority, term and effectiveness status.

Additionally, we classified the policy texts. On the basis of several index systems, EC can be classified as linked to environment, economy, resources, society and culture [9,12,22,28]. Based on the *Indicators of National Ecological Civilization Construction Demonstration District* issued by the China Ministry of Ecology and Environment in 2019, we divided the policy documents into seven categories: ecological economy (EE), ecological life (EL), ecological culture (ECU), ecological institution (EI), ecological safety (ES), ecological space (ESP), and ecological synthesis (ESY) (see Table 1).

### 2.2. Data Analysis

Co-word analysis, coauthor analysis and cluster analysis were used in this study. The effectiveness of these methods has been confirmed in previous studies of policy documents [49,50,51,52]. Co-word analysis, which uses the keywords in documents to build relationships, is commonly applied to identify hot spots or trace the main development directions in policy [50,53,54]. The co-word frequency of keywords is applied to measure degree centrality, which shows the strength of the relationships among high-frequency items [49], and is also used to show the size of every keyword node [52]. Furthermore, for authors, that is, the policy-making departments, a similar method known as coauthor analysis can be used to explore the relationships among policy-making agencies [51,55]. Moreover, the outcomes of co-word analysis and coauthor analysis measure the clusters that collect the most connected keywords into one group and exclude others [56]. Thus, the main topics and development path can be easily understood.

The specific procedures are as follows:

For the keywords in policy documents: first, by reading policy documents and related relevant materials, we collected and organized words related to ecological civilization, environmental conservation and government agencies to build the custom thesaurus. At the same time, we constructed the filter thesaurus to eliminate adverbs, meaningless verbs and irrelevant nouns. Second, we used the Content Mining System User Manual Version 6.0 (ROSTCM6), a content mining system designed by Wuhan University in China, to segment sentences into words as the keywords. In this process, the custom thesaurus and the filter thesaurus were applied to ensure the accuracy of the results. Third, we counted the frequency of each keyword extracted from the segmentation results and used ROSTCM6 to establish the co-word matrix of different categories of policies in different phases. Next, we employed multidimensional scaling (MDS) and the “Netdraw” function of Ucinet 6.0 to visualize and lay out the co-word network and delete some meaningless words in the network. Fifth, the network data were imported into Vosviewer 1.6 for network visualization, and subgroups were marked with different colors.

To analyze organizations and functions: in data preprocessing, we have categorized the topics of the policy text according to the concrete contents. Every document can represent the function or goal of one or more organizations in terms of EC. Based on this, the researchers analyzed the policy-making agencies and policy functions in each document and constructed a 2-mode organization-functional subordinate network. Furthermore, Ucinet 6.0 was used to transform the 2-mode network into two 1-mode networks: the policy-making agencies network and the policy functions network. Then, we calculated the normal degree centrality, which would show the policy priorities in 1-mode networks, and betweenness centrality, which would indicate organizations’ coactions in 2-mode networks for each policy goal in different time quanta. In addition, we measured the network density and average path length of the organizational network to further illustrate the variation in inter-organizational relationships.

## 3. Results

Figure 1 illustrates the number of EC policies in Fujian Province issued annually from 2004 to 2020. In our research, we segment the phases by factors from two perspectives when there is no official division of stages for the construction of EC as a reference. First, there are two clear temporal turning points in the growth trend of policies number, 2013 and 2018. The dataset contains 1497 policy documents over 17 years. From 2004 to 2012, the number of EC policies showed slow growth. Starting in 2013, the number of new EC policies climbed drastically, peaking at 316 documents in 2017, nearly sixfold the number in 2012. After the peak, the number decreased rapidly beginning in 2018. Second, these two points in time coincide with two core events in China’s central environmental protection policy decision-making. One of these, as stated in the “Making Great Efforts to Promote Ecological Progress (MGEPEP)” section of Hu’s report at the 18th Party Congress on 8 November 2012, is that EC construction would be part of the national development frame, gaining tremendous driving power from politics. Since this event took place at the end of 2012, it would mainly have an impact from 2013. Another is the “Amendment to the Constitution of the People’s Republic of China” passed by the 13th National People’s Congress, which gave EC higher legal status and a stronger guarantee of political power. This event occurred in the first half of 2018 and would therefore also have a large impact on 2018. In sum, based on the changing trends and quantities of EC policy combined with the key events of China’s EC actions, the policies can be divided into three stages: Phase I extended from 2004 to 2012, with preliminary exploration; Phase II lasted from 2013 to 2017 and developed quickly with the support of central policy-making; and Phase III started in 2018, ended in 2020 for this study, and would continue for a long time.

Figure 2 demonstrates the overall organization-functional network. The circular nodes represent policy-making agencies, and the square nodes represent policy goals, which can also be considered functions. This network shows the relationships between EC policy-makers and policy functions. The larger the circles, the higher the coordination, the more positive the response, and the stronger the power of the agencies; the larger the squares, the more significant the functions. Corresponding to different periods, the numbers of policy goals and key goals change. The numbers are 19, 20, and 20 for Phases I, II, and III, respectively. We calculated the normal degree centrality of policy goals to identify the key goals of these stages (Table 2 shows the top 10 function values) and network density and betweenness centrality (Table 3 shows the top 5 goals) to clarify organizational relationships.

### 3.1. Phase I: Early Exploration (2004–2012)

During the early period of EC construction in Fujian Province, 218 EC policies were issued from 2004 to 2012. Figure 3 illustrates the main policy priorities: ecological conservation, pollution discharge treatment, comprehensive improvement of urban and rural human settlements, ecological assessment and supervision, policy support, promotion of green breeding techniques and development of green industry (especially agriculture and tourism). EP, RS and EQI occupied the top three positions, meaning they were the core of policy-making. RCS was not a concern in this stage.

The clustering network of policy keywords in this period mainly concerned two aspects, human settlements and ecological governance, and green industrial development was also considered. Using an “administrative committee” to “lead off” and be in “charge”, which meant subdividing the responsibility for environmental improvement among grassroots management units, was the main way to transform urban and rural environments. This approach was applied in “greening”, “garbage disposal” and “health”. Regarding environmental governance, “regulation” and “assessment” with the government as the subject were the main policy tools, aimed mainly at enterprises. In addition, some governmental investment and technology promotion were added to policies to support industrial development in the direction of low pollution and low energy consumption. Specifically, although CC as a policy goal showed a high normal degree centrality, there was no relevant subgroup in the clustering network.

A total of 67 organizations were involved in the development of EC policies during this stage. The organizational network density was 0.581, and an average of 1.419 departments had to span two organizations to achieve coordinated communication. In this 2-mode network, the policy goals that the departments jointly promoted were mainly related to “culture” (including CC and PGC), while there was less willingness to jointly promote other policy goals in terms of common actions. The actual cost of paying for culture building is low, and there are fewer factors to consider. Organizations can enact a cultural advocacy policy at almost no cost. The core departments in the organization–functional network were the Fujian Provincial Committee of the CPC (FJPCCPC), the Fujian provincial and municipal governments, and the Fujian Provincial Bureau of Ocean and Fisheries (FJPBOF).

### 3.2. Phase II: Rapid Growth (2013–2017)

In this phase, EC developed at a rapid pace, with 913 policy documents, accounting for 62.1% of all policy documents over the 17-year period. The EC policies focused on ecological restoration, ocean and fishery issues, pollution monitoring, policy support, publicity, pilot evaluation, green building promotion and land planning (see Figure 4). GI replaced EQI and became one of the top three. PD also climbed in the rankings.

In this period, the EC policy documents became more detailed. Forest and ocean issues were divided into different policy directions. Measures on forest systems tended toward ecological restoration and aimed to improve the ecological environmental quality of forest regions by “afforestation” and “protected areas”; measures for marine systems emphasized the utilization of resources and regulations related to the development of sustainable marine fisheries by “law enforcement”. After the improvement of the living environment and the promotion of agricultural techniques, the main policy direction became the promotion of green building techniques and the further optimization of urban land planning. The government no longer considered “supervision” and “regulation” its first functions, and “service” became the main feature of its role, which was an important embodiment of China’s service-oriented government construction in EC. Although it was not reflected in the centrality ranking of policy functions, CC constituted a subgroup with high centrality at this stage. Creating a public cultural “atmosphere” became a significant social foundational support for the construction of EC. Specific ecological pilot policies were also carried out with attached “norm” and “evaluation” aspects to confirm whether promotion was feasible.

The number of organizations in the network proliferated to 120. At the same time, the density of the organizational network decreased to 0.556, and the average path length increased to 1.444. This meant that the speed of information exchange and information transmission between organizations declined, and there were some obstacles to inter-organizational collaboration. When the density of the organization–functional network gained strength (from 0.196 to 0.225), the overall betweenness centrality of policy goals other than CC goals increased. The change in density meant a promotion in the power exercising process. This indicated that the power exercise process was optimized, and more organizations were willing to try to take on more functions to jointly promote a particular policy goal. The Fujian Provincial Department of Development and Reform Commission, which is a macro-regulatory department for integrated research and the elaboration of economic and social development policies, was added to the core departments that existed in the previous phase. The support of the central policy allowed a large number of organizations to join, and just such a macro-regulatory department was needed to prevent disorderly development.

### 3.3. Phase III: Deepening Stage (2018–2020)

There were 362 EC policy documents in this phase, and the problems focused on were ecological law enforcement, policy support, publicity and supervision, cadre training, pollution control, farmland protection and domestic waste disposal (see Figure 5). EP and RS maintained their roles, while PD became significant and CV withdrew from the ranks of important policy goals. Specifically, compared with the former two stages, the values of all policy functions’ degree centrality continued to decrease, which signified that the relationship strength among different goals was diminishing. This change manifested a loosening integration of specific separated ecological goals, which meant that subentry purposes were entering a deepening development stage in specific areas rather than cooperation as a whole. The overall framework seemed to have been built flawlessly, and it was time to thoroughly improve the details.

The policy themes covered in this period did not undergo obvious changes compared with those in the previous stages, but the policy goals became more focused. As EC was incorporated into the constitution, accountability, expressed by terms such as “legal”, “law enforcement”, “charge” and “responsibility”, was highlighted. The construction of EC became more legalized overall. Grassroots management was strengthened in this phase, and both forest management and land planning showed characteristics of “territorial management”. In terms of publicity and education, the education and training of cadres of the CPC formed a separate subgroup; the role of public EC supervision began to be emphasized in public awareness campaigns. As the density of the organization–functional network further increased (from 0.225 to 0.301), policy-makers seemed wiser and more on target. The government’s policy support was no longer a single “fund” support but a more systematic “banking” support, and the policy status of “supervision” was strengthened. “Data” and “index” showed that EC evaluation in Fujian Province had become more scientific and practical. Interestingly, though Fujian Province has been concerned about the arable land issue since 2002 and basically achieved a balanced occupation of farmland, there were many nodes concerning cultivated land during this period. Some other issues, such as domestic garbage treatment, also appeared in this stage, indicating that the policies had begun to focus on specific problems rather than overall construction. In this stage, there was no need to make major changes to the overall framework, but partial amendments were needed to solve specific aspects of those problems that still existed.

In Phase III, the number of organizations involved in EC policy-making decreased to 97. The organizational network density increased to 0.593, and the average path length decreased to 1.407. With the withdrawal of some marginal organizations, communication barriers were removed from organizations in the network, and communication costs fell. While barriers had been reduced, cooperation among organizations had not been enhanced. The overall betweenness centrality of policy goals declined, even below that of the first stage, which might have been related to differences in EC construction goals set by the central government in different eras. The decentralizing organization ecosystem indicates a tendency of devolution and independence that granted the subject more independent rights and a more professional division of functions. As a new member of the core sectors, the Standing Committee of the People’s Congress of Fujian Province enhanced its legislative role after EC was written into the constitution. Legislation made the policy content of EC construction more standardized and raised the threshold of participation.

## 4. Discussion

Based on policy quantitative analysis results, in this section, we explore and analyze the characteristics and basic patterns of China’s evolving EC construction policies from the perspectives of targets, actors and institutions (see Figure 6).

### 4.1. Target Setting

The target of EC construction was set in the *Outline of the Overall Plan for the Ecological Province Construction* created by Fujian Province, which was a special kind of policy document for China. Documents related to “planning”, such as the Five-Year Plan, are the basic method for China’s socialist modernization and governance. This provides a structural and macroscopic strategy for the market and establishes the basis of public service duties for the government [57]. Due to the specific period, the Five-Year Plan showed distinctive traits in terms of target setting, strategy mapping and reform launching [58,59]. Therefore, in energy and environmental studies, it is often used to limit the time span, measure management performance and predict potential development [60,61,62]. The target setting of the planning was based mainly on two questions: “What goals should we achieve?” and “When should we finish the tasks?” Based on these two questions and the research results, we suggest that the policy actions of EC construction took planning as the main line and were affected by the tenure of the leading officer and major central government policy decisions. The planning guide was significant in the process, while the action strategy was transformed under the influence of changes in the political, social and economic environments. There was a gap between planning target setting and actual policy actions over time, which may have been affected by official tenure and central government guidance. In local planning, the provincial government brought its subjectivity into play. In its planning, Fujian Province applied a different strategy than that of the normal central Five-Year Plan regarding the time limit: the time allowed was longer, and the segment criterion was not the regular five years. In the plan made by the provincial government, the task of constructing the ecological province was divided into three phases: the first phase (before 2005), the second phase (2006–2010) and the third phase (2011–2020). The first phase cannot be seen as a complete construction stage. Because only one year was allocated for the “big goal”, this seems more like a phase for resource integration and preparation to “start” EC construction. Our results suggest that EC actions in Fujian Province can be segmented into three phases that cover different time intervals in terms of planning. The reasons why the time span differences occurred are as follows. First, the tenure and turnover of environmental protection officials significantly affected government environmental protection actions [63,64]. Xi Jinping was the official who led this planning while he was the governor of Fujian Province. Before the planning officially took effect, Xi was reassigned to Zhejiang Province. As a result, until he was elected president of China in 2012, the progress of EC construction went very slowly, as indicated by the annual number of EC policy documents. During his tenure in Zhejiang Province, he proposed EC system construction measures such as “natural recuperation”, division of ecological function zones and “ecological compensation”, which formed his initial theoretical system of EC. On 15 December 2012, the same year as MGEPEP, Xi came to power. In the next year, EC was incorporated into the “five-in-one” blueprint for the cause of socialism with Chinese characteristics, which meant that EC became one of the main governing concepts. Since then, EC entered a rapid development stage and the *Integrated Reform Plan for Promoting Ecological Progress* introduced in 2015 further accelerated its development. Second, the guidance of the central government is significant in local governance. Because of supervision, evaluation and other regulations from the central government, the process of implementing planning goals is constrained [57]. Local government must revise its policy actions according to central decision-making, which may disrupt the original arrangement. The annual number of policy documents (Figure 1) can be seen as the government’s EC construction activity, which follows the weightier events led by the central government, such as the constitutional amendment. The center stipulated the general direction and used performance assessment pressure to urge local government to revise its policy actions. After the beginning of the third phase, EC construction of Fujian Province turned from planning to actual policy adjustment, causing a discrepancy between actual policy actions and planning in terms of time.

There was a strategic change in government action focus, showing a trend from broad construction to problem refinement. From the time that EC was mentioned in the first policy document, Fujian Province tried to structure the framework for constructing the EC province. Planning and deployment showed high centrality in the policy goals, indicating that EC was deemed a comprehensive program. From Phase I to Phase II, policy-makers tried to extend the policy coverage from industry to the living environment. An increasing number of aspects of the ecological environment were included. As Fujian is a coastal province with the highest rate of forest coverage and ocean, forests and oceans became high-priority issues, matching industries such as agriculture and tourism. In Phase III, the policies gradually turned to specific problems. The marine environment was no longer the main topic, while forests needed fewer network ties. Land use, especially for cultivated land, became an important subject. Similarly, garbage disposal gained importance in the network. The framework construction was continuous, but its focus changed to specific problems. This process showed features of refinement. First, the items of focus developed in detail. Initially, the forests and ocean were considered the same environmental issue without differentiation. However, they were quickly separated, and subgroups appeared in their individual networks. Similarly, publicity was separated into public and cadres. Second, the functions changed to become clearer. For the government, responsibility coverage developed from simple regulation to comprehensive services and evaluation, including data collection and index setting. Power also underwent refinement, which can be seen as a kind of power decentralization or division. Responsibility for EC construction was distributed to grassroots units, and the tremendous and complex task was broken down into small goals that most local officials could understand and achieve through “territorial management” or “localization management”. Regarding citizens, they were empowered by new rights. However, there was no obvious change in the role of enterprises.

### 4.2. Actors’ Functions

The construction of China’s EC was the result of the combined efforts of different participants, including the CPC, the government, citizens and the market. In the three phases, the roles of some actors changed considerably, while others remained almost the same.

The CPC was the core leader and decision-maker throughout the EC construction process. At the beginning of EC construction, the planning was performed and stated by the FJPCCPC, which is the highest authority of Fujian Province. In a specific policy-making process, FJPCCPC, committees of the CPC of the cities of Fujian Province and their branches played a leading role. As environmental actors, they issued a large number of policies. Furthermore, the People’s Government of provincial and local governments published core documents in conjunction with them to guarantee policy effectiveness. The existence of party regulations also made the CPC the main supervisor and guarantor of EC construction actions. In addition to formal laws, CPC regulations are the most direct constraint on environmental governance officials. The CPC’s unscheduled environmental inspections and lifetime accountability for environmental governance put enormous internal pressure on environmental governance officials, urging them to do a better job of ecological governance. Therefore, the leadership of the party was the essential feature of EC construction.

The government’s role changed from that of a manager to that of a service provider. In the early stage, China’s practice of EC construction started as a mode of government domination, as shown in forcible regulation and direct financial support. After the notion of EC was proposed, a variety of policies were issued to construct the EC framework. Figure 3 shows that all the actions were related to government force, while enterprises were regulated objects. In addition, there were no subjects related to citizens or other social organizations such as the media. Although some documents mentioned public awareness campaigns for citizens, the government deemed the public to be a passive receiver of information to be educated and taught. There was no governance network but rather a single-direction producer-to-receiver chain. However, governance for sustainable development requires deliberative governance and the involvement of societal actors [65]. Thus, China’s government is trying to transform the functions of multiple actors. The government is converting its image of a manager to that of a service provider. Figure 3, Figure 4 and Figure 5, focusing on policy support, show how “service” emerged and grew. In Phase III, the “revolution” indicates a possible deeper transition of the government’s role. Interestingly, the mode of service changed in the third stage; that is, “finance” and “fund” support transitioned into comprehensive “banking” support. For officials, the government strengthened education and training to improve cadres’ “capacity” for environmental governance.

When the perspective shifts to social and market players, the change mainly occurred among citizens. Against the background of the Information Age, it is critical to rethink the role of citizens [66]. “Supervision” appeared in the publicity subgroup, possibly due to the transition of the public’s role. The publicity receiver became the supervisor and even a participant in policy-making. In addition, some “institutions” were engaged in the policy support subgroup. It is possible that this change stemmed from the government’s ecological culture strategy, as President Xi Jinping said, “The construction of ecological civilization is a revolutionary change involving the mode of production, way of life, way of thinking and values”, which needs mass participation. People orientation is gradually becoming one of the main characteristics of China’s EC construction [27]. It has become an important political responsibility of the government to continuously improve the quality of the environment and provide better ecological public goods. To achieve this goal, citizens’ ability to express their environmental rights and interests, their willingness to actively participate in government decision-making, and the social supervision of environmental governance are necessary. In terms of the market, we found no obvious change in any network. The market remained the main object of governmental regulation. However, we found that the specific regulatory content changed from mere limitation and restrictions to encouraging companies to self-regulate. The government was trying to mobilize business ownership in environmental protection.

### 4.3. Institutional Guarantees

Environmental legislation has been established to resolve conflicts over environmental resources and to make the community and market subjects accountable for environmental damage [67]. These laws can function as a legal basis, tendency guidance and framework restriction in policy instrument design and selection [68,69]. Chinese EC construction has moved into an era of legalization, as it is embedded in the national political framework and legal system. Regarding renewable energy, clean production, environmental impact assessment and pollution control, China has issued a series of the most advanced laws in the world [70]. The construction of EC started from a conventional government policy document, and the main guarantee was administrative power. In this paper, we show that the centrality of keywords related to law and its execution rose continually. Combined with the legislative process of Chinese environmental laws, we suggest that the EC was developing into more of a force-of-law stage.

Tracing the latest legal action since EC was integrated into the constitution, we confirmed that laws on EC became more detailed, more operational and more rigorous. After the amendment of the constitution, the Chinese government issued specific laws, such as the *Resource Tax Law (2019)*, *Biosecurity Law (2020)* and *Yangtze River Protection Law (2020)*, to fill the gaps in special fields. Around these events, more than 18 laws on the environment were reformulated, including the *Civil Law (2020)*, *Criminal Law (2020)* and the latest *Administrative Law (2021)*. In particular, the *Criminal Law* increased the criminal files of environmental destruction crimes, increased the legal punishment, and criminalized “fraud” related to environmental assessment and monitoring. Fujian Province, as an outstanding district in EC construction, also took the lead in introducing the law on the chief river system (2019). These efforts to perfect the environmental law system have created the possibility of new policy instrument usage in ecological conservation for governments that want to guide and correct social conduct related to the environment [71]. As shown in Figure 5, this policy trend emerged in forest protection, which was probably a consequence of the revision of the Forest Act (2019) and the fact that Fujian has the highest forest coverage in China. Not only do keywords such as “legal” and “law enforcement” show high centrality, but the supporting administrative measures, including “charge” and “regulation”, are also significant in this network.

Although the legal basis is being consolidated, there are still obvious problems in the efficiency of law enforcement. The legislative specialization of environmental laws has aimed to fill gaps in particular ecological aspects, but has failed to coordinate interdepartmental relationships. The policy networks show that the government is scientifically overhauling the performance evaluation system, which can be inferred from terms such as “standard”, “data”, and “index”. The policies on performance evaluation of EC construction have created effective incentives for local governments, especially in encouraging officials to make ecological conservation one of their political priorities [16]. However, as a systematic process, EC construction needs cooperation among different agencies, while China’s environmental protection is facing “jiu long zhi shui”, which means several departments managing the same issue with different opinions, leading to the failure of governance. Both the laws and the policies emphasize accountability, while multisectoral linkage has not shown significance. At the same time, legalization has raised the threshold for organizations to participate in EC decision-making, which will potentially exclude those participants who are professionally capable of solving EC construction problems. China’s EC has reached a specific legalization status, but there is still a great necessity for operational laws and the reform of coordination.

### 4.4. Policy Implications

For worldwide sustainable development and environmental problem solutions, the process of Chinese EC construction may provide insight as follows: (1) It is necessary to build a framework and make planning a priority in environmental governance. A sound blueprint can reduce policy bias and allow appropriate policy adjustments to be made. (2) Relying only on administrative power is not a wise way to effectively promote ecological governance. The regulations of the CPC and strict environmental law construction must be steadfastly promoted. (3) Although the solution to environmental problems needs to be driven by the government, the people are the mainstay of the process. Environmental policy-makers must listen to the demands and interests of citizens in addition to considering economic and social environmental changes. Although there are differences between Chinese and European regions and other districts’ modes or styles of ecological modernization [72], certain experiences, including policy planning, policy goal transitions, people-oriented policy-making and responsibility thinning, may serve as a basis for exploring more effective EC globally.

## 5. Conclusions

Because of increasingly serious environmental problems, China promoted the notion of EC to guide environmental governance and sustainable development. EC construction is significantly different from traditional environmental protection or sustainable development. It has been distinctly political from its birth and has been used as an ideological framework by the Chinese government, not only in terms of its role in the environment, but also in influencing other areas of governance in China. Aiming to present China’s EC practices, this research employed co-word analysis, coauthor analysis and cluster analysis to describe and visualize the transition of policy priorities on the basis of policy keywords and organization–functional networks in three stages. We found that there are three core elements of EC: target setting, actors’ functions and institutional guarantees. In addition, there are certain important characteristics: planning guidance is significant when actual policy actions transition to promoting issue refinement from frame construction, and planning implementation is deeply influenced by leadership tenure and central decisions; the subjects participating in EC transition from government-led to multiple actors and the transformation of their functions; and the construction process’s development from policy support to specific legal guarantees. Nevertheless, this paper has limitations, including some missing policies due to policy documents that were not uploaded to official websites or the policy database. Additionally, the cooperation among government agencies and social organizations is more complex than we describe in this paper, and the co-word network can show only the most frequent keywords of policy documents. These limitations will inform the directions of our future research efforts.

## Figures and Tables

**Figure 1 ijerph-19-08627-f001:**
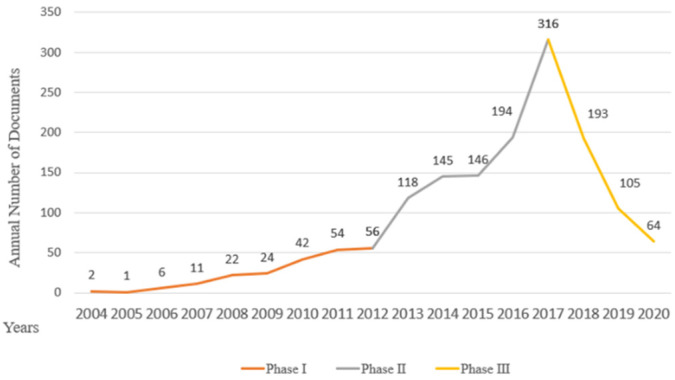
Annual number of documents.

**Figure 2 ijerph-19-08627-f002:**
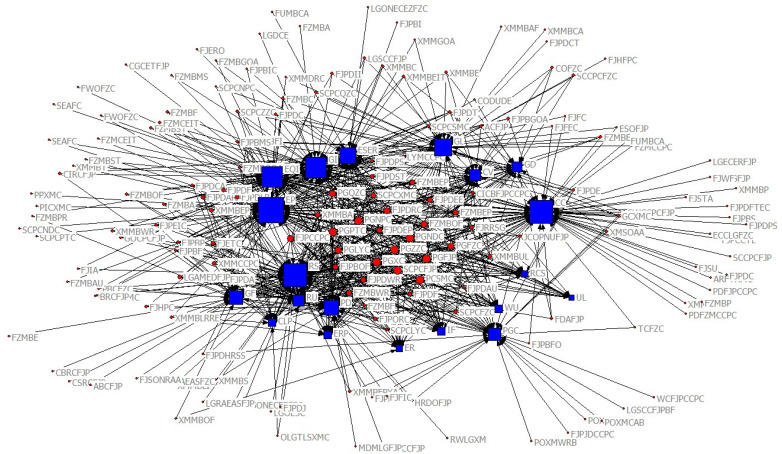
Overall organization-functional network (Phases I to III).

**Figure 3 ijerph-19-08627-f003:**
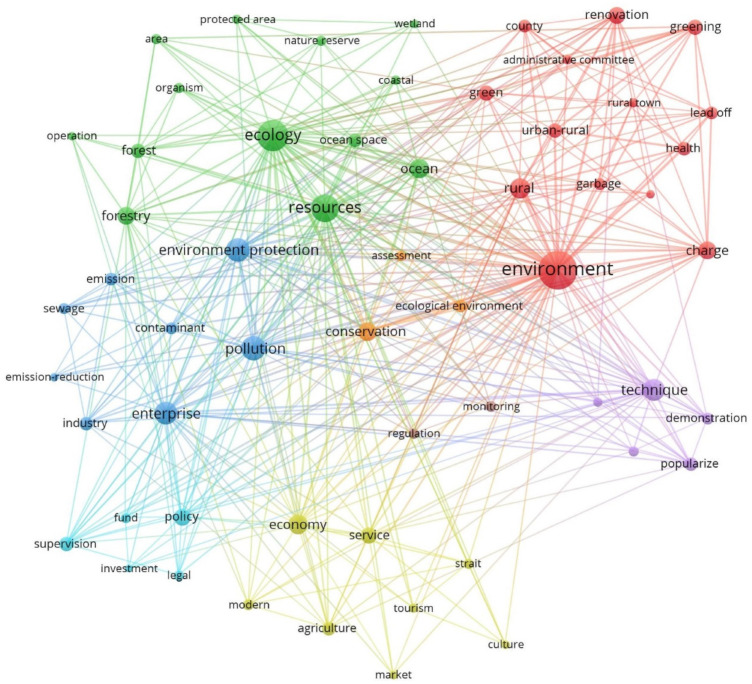
Co-word analysis and cluster analysis of keywords (2004–2012).

**Figure 4 ijerph-19-08627-f004:**
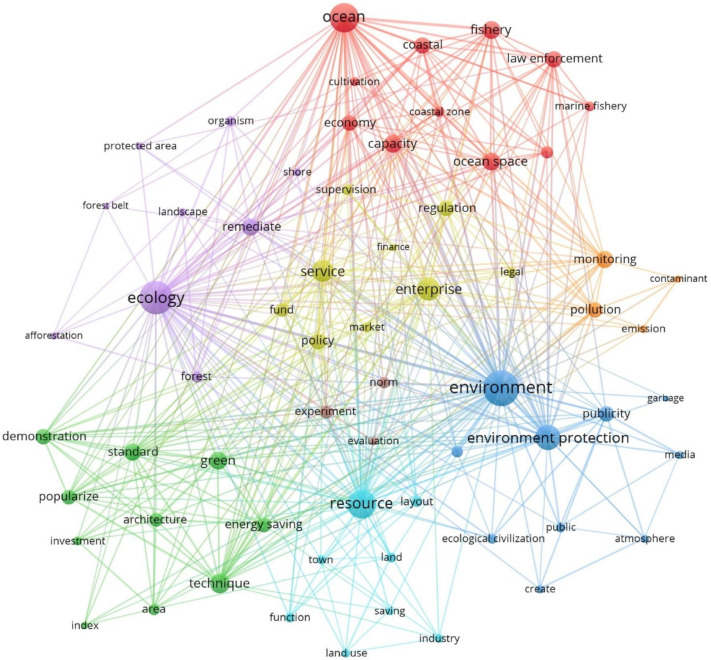
Co-word analysis and cluster analysis of keywords (2013–2017).

**Figure 5 ijerph-19-08627-f005:**
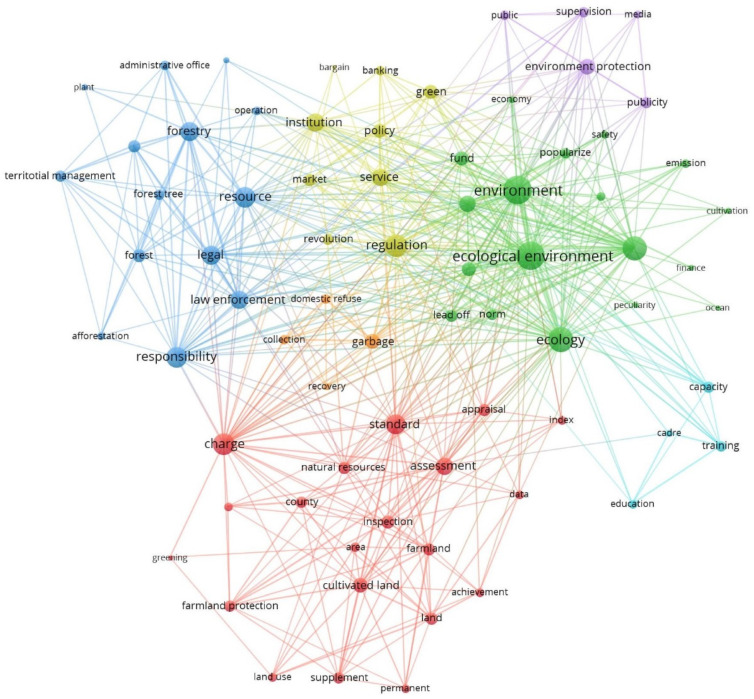
Co-word analysis and cluster analysis of keywords (2018–2020).

**Figure 6 ijerph-19-08627-f006:**
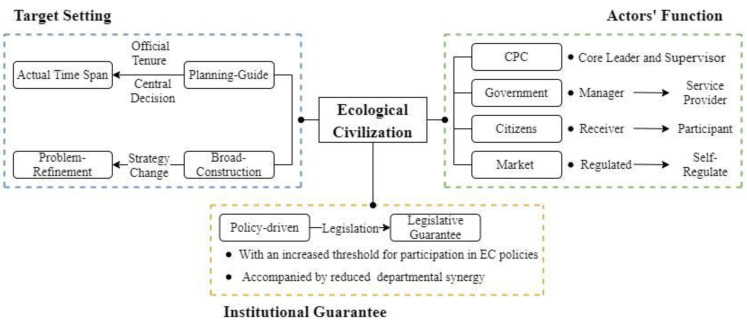
Core elements of EC and their pathways.

**Table 1 ijerph-19-08627-t001:** Policy document classification.

Category	Policy Goals
Ecological Economy (EE)	Energy Saving and Emission Reduction (ESER)
	Green Industry (GI)
	Infrastructure (IF)
	Green Finance (GF)
Ecological Life (EL)	City Virescence (CV)
	Garbage Disposal (GD)
	Green Lifestyle (GL)
	Water Usage (WU)
Ecological Culture (EC)	Party and Government Culture (PGC)
	Citizen Culture (CC)
Ecological Institution (EI)	River Chief System (RCS)
	Responsibility and Supervision (RS)
	Planning and Deployment (PD)
Ecological Safety (ES)	Environmental Quality Improvement (EQI)
	Ecosystem Protection (EP)
	Environmental Risk Prevention (ERP)
Ecological Space (ESP)	Urban Layout (UL)
	Cultivated Land Planning (CLP)
	Ecological Redline (ER)
	Resources Utilization (RU)
Ecological Synthesis (ESY)	Including more than 2 different categories’ policy goals

**Table 2 ijerph-19-08627-t002:** Normal degree centrality of policy goals.

Rank	Overall	Phase I	Phase II	Phase III
1	EP (0.364)	EP (0.454)	EP (0.343)	EP (0.325)
2	RS (0.284)	RS (0.392)	RS (0.295)	RS (0.221)
3	PD (0.243)	EQI (0.372)	GI (0.264)	PD (0.181)
4	EQI (0.232)	GI (0.366)	PD (0.244)	GI (0.157)
5	GI (0.230)	CC (0.350)	EQI (0.239)	EQI (0.151)
6	RU (0.163)	PD (0.298)	RU (0.151)	RU (0.124)
7	ESER (0.120)	CV (0.262)	GF (0.108)	ESER (0.099)
8	GF (0.107)	GL (0.250)	ESER (0.102)	GF (0.075)
9	CV (0.098)	RU (0.205)	IF (0.101)	GD (0.075)
10	CC (0.089)	ERP (0.189)	CV (0.098)	GL (0.064)

**Table 3 ijerph-19-08627-t003:** Betweenness centrality of policy goals in the 2-mode network.

Rank	Overall	Phase I	Phase II	Phase III
1	CC (0.248)	CC (0.457)	CC (0.243)	EP (0.254)
2	EP (0.173)	PGC (0.150)	GI (0.177)	EQI (0.160)
3	RS (0.172)	EP (0.106)	EP (0.136)	RS (0.146)
4	EQI (0.118)	GI (0.081)	EQI (0.130)	CC (0.111)
5	GI (0.110)	GL (0.068)	RS (0.122)	GL (0.095)

## Data Availability

The data presented in this study are available on request from the corresponding author. The data are not publicly available due to the fact that the data are also being used in an ongoing study.

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
