# Peer review of "How Has China Structured Its Ecological Governance Policy System?—A Case from Fujian Province"

_ijerph, 2022, doi:10.3390/ijerph19148627_

Round 1

Reviewer 1 Report

Well done.

Author Response

Comments 1: Well done.

Response: Thank you very much for your recognition of our research.

Reviewer 2 Report

The manuscript is well written and well presented. The work does not present any particular novelty but is nevertheless appreciable for the detail of the approach followed.

The "Data sources and preprocessing" paragraph can be better explained especially the part of the data analysis.

Moreover, he paper could be made more complete and exhaustive through some literature comparisons of similar analyzes in other geographic areas thus making the presentation of the results clearer.

In general it is still a good work which, in my opinion, can be published after minor revison as suggested.

Author Response

Comment 1: The manuscript is well written and well presented. The work does not present any particular novelty but is nevertheless appreciable for the detail of the approach followed.

Response: Thanks for your recognition of our research. We will pay attention to the issue of novelty in our future research.

Comment 2: The "Data sources and preprocessing" paragraph can be better explained especially the part of the data analysis. followed.

Response: We completely agree, and we made a revision following your suggestion. We have adjusted and optimized this section.

Comment 3: Moreover, the paper could be made completer and more exhaustive through some literature comparisons of similar analyzes in other geographic areas thus making the presentation of the results clearer.

Response: Thank you for this very useful comment. In our new version, we add more literature of similar analyzes in other areas to make the results presentation clearer.

This manuscript is a resubmission of an earlier submission. The following is a list of the peer review reports and author responses from that submission.

Round 1

Reviewer 1 Report

EC is a new global environmental governance system to benefit people proposed by China. It is important to figure out the process and performance of the policy. The study takes Fujian Province as a case study and reviews the EC construction system by systematic analysis of policy documents from 2004 to 2020. Overall the research design is appropriate and the result presentation is clear. Here are several suggestions for improvement.

Firstly, in the result part, the EC is divided in 3 phases. How is the delineation of ecological civilization construction phases determined? It is unclear.

L27-28, The statement is a bit too absolute, China's rapid economic development is often accompanied by high pollution and high energy consumption, but not all.

L61-108, The transition between the two paragraphs is rather confusing and lacks logic.

L312, The description Interestingly, the policies began to pay attention to the treatment of cultivated land and domestic garbage during this periodmaybe wrong. Actually, in 2002, Fujian Province has implemented the first round of policy of returning farmland to forest and grass, and in 2007, the State Council issued the Notice of the State Council on Improving the Policy of Returning Farmland to Forest to further strengthen the consolidation of the results of returning farmland to forest, and from 2010 to 2012, Fujian Province basically achieved a balanced occupation of farmland.

L365-372, It is not clear that why EC became one of the main governing concepts of China under his advocacy and entered a rapid development stage after Xi came to power.

L365-382, although the reasons for the formation of three stages of EC in Fujian Province are explained, all three reasons seem to explain only the slow development of the first stage of EC, and do not explain the rapid development of the second stage and the deep development of the third stage.

Reviewer 2 Report

Review of the manuscript

„How has China structured its ecological governance policy system? ——A comprehensive review of ecological civilization construction”

for the

Given the title of this manuscript, it promises to deal with an interesting topic. It is supposed to be about the concept of "ecological civilization", which serves as an attempt in China to get a grip on the immense environmental problems that arise as unintended side effects of continued industrialization. In fact, a "comprehensive review" of the construction of "ecological civilization" is promised. The abstract tells the reader that this "review" will be carried out by means of a bibliometric analysis of policy documents. Furthermore, the reader learns that the authors of the manuscript have obviously worked out a case study, namely in Fujian Province.

Further reading quickly cools down the enthusiasm for the topic and the interest in the methodical (and methodological) challenges. Unfortunately, the manuscript has four fundamental deficiencies that, in my opinion, prevent its publication.

(1.)  Although the title promises statements about China as a whole, the authors - as already mentioned - elaborate a case study and build up an argumentation according to which the whole country is inferred from the analysis of one part. This is, in principle, possible if the basic rules for generalizing the findings developed on the case are taken into account. This requires highlighting what is special about the case and showing how the study region relates to all other regions in China. Unfortunately, the authors do not take this approach. They argue by referring to statements made by the Chinese President and Xinhua News Agency to show that Fujian is a special region (p. 3, lines 117-124). Yet this does not help to assess the geographic and sociocultural characteristics that distinguish this region from the rest of China. The fact that the authors do not address some of the region's distinctive features until p. 11, in the chapter discussing the research findings (instead of including these features in their assessment of the policy documents), raises fundamental questions about the extent to which the methodology of case-study-based reasoning was even considered in writing the manuscript.

(2.)  This brings me to the second fundamental flaw. The authors make statements about the characteristics and basic patterns of the construction of what is called "ecological civilization" on the basis of bibliometric analyses. The data analysis procedures used are structure-discovering procedures. They are not designed to test hypotheses and therefore do not permit firm conclusions about relationships or even cause-and-effect relationships. Authors must take this seriously into account in their argumentation. This means that they can actually only discuss arguments of plausibility as to which factors have generated the development of the data situation as shown in figure 1. This sort of balancing is not undertaken in the manuscript. The authors simply state that, viewed over time, changes in the data situation have resulted incrementally from domestic political developments in China. However, what influence did foreign policy events have on developments in Fujian, for example, and what local and regional developments were relevant? The authors say nothing about this. It may be plausible to attribute the changes in the frequencies and structure of the co-words and keywords to the domestic political events mentioned. But the authors need to provide evidence for this, ideally in a separate chapter that explains the process of environmental policy in China's provinces before the data analysis and discusses the significance of the policy documents studied.

(3.)  This leads me to the third fundamental point of criticism. The authors discuss their sample construction highly insufficiently. They state "The policy document data investigated in this paper were derived from government 125 official websites and the PKULaw Database (the largest Chinese policy database, which is 126 continuously updated with the most recent documents)" (p. 3, lines 125-127). Next, they describe how they constructed the selection by searching for the particular keyword. But key questions remain unanswered. First and foremost among these is the question of what kind of documents they are actually talking about. The term "policy documents" is not very specific. Are they concept papers, strategy papers, administrative orders, or entirely different texts? Or are there various functional texts in the databases mentioned and thus also in the sample? What is the significance of these documents for which actors in the political process? Here, the authors of the manuscript would have to ensure clarity and transparency and, if necessary, conduct the analyses for individual document genres. Perhaps colleagues familiar with Chinese environmental policy will be perfectly clear how to respond to these questions. However, in an international journal, this knowledge cannot be assumed.

(4.)  It is essential to have a common frame of reference for a study like this one in order to gain insights into environmental policy in international comparative exchange. Unfortunately, such a frame cannot be found in the manuscript. The distinctive feature of the policy of "ecological civilization" would have to be distinguished from the concept of "nature conservation" referred to in the text, but also from the "ecological modernization" briefly mentioned at the end. To what extent does "ecological civilization" lead to sustainable development? What is the underlying understanding of sustainability? There would be various possibilities to name points of comparison and to develop concepts for this purpose. Without such a frame of reference, empirical data and analyses are blind, even if data are analysed in an exploratory way.

In sum, these points of criticism present such a serious problem that a publication of the data analyses can only be recommended after a fundamental reworking of the text. It is doubtful whether the authors can do this on their own. Someone from the authors' circle should have noticed the fundamental deficiencies during the preparation of the present manuscript. Against this background, I cannot decide on the recommendation to review the manuscript again after a fundamental revision.